# Spatial Distortion Assessments of a Low-Cost Laboratory and Field Hyperspectral Imaging System

**DOI:** 10.3390/s19194267

**Published:** 2019-10-01

**Authors:** Andrija Krtalić, Vanja Miljković, Dubravko Gajski, Ivan Racetin

**Affiliations:** 1Faculty of Geodesy, University of Zagreb, 10000 Zagreb, Croatia; 2Faculty of Civil Engineering, Architecture and Geodesy, University of Split, 21000 Split, Croatia

**Keywords:** imaging spectrometer, pushbroom scanner, spatial calibration, MTF

## Abstract

This article describes the adaptation of an existing aerial hyperspectral imaging system in a low-cost setup for collecting hyperspectral data in laboratory and field environment and spatial distortion assessments. The imaging spectrometer system consists of an ImSpector V9 hyperspectral pushbroom scanner, PixelFly high performance digital CCD camera, and a subsystem for navigation, position determination and orientation of the system in space, a sensor bracket and control system. The main objective of the paper is to present the system, with all its limitations, and a spatial calibration method. The results of spatial calibration and calculation of modulation transfer function (MTF) are reported along with examples of images collected and potential uses in agronomy. The distortion value rises drastically at the edges of the image in the near-infrared segment, while the results of MTF calculation showed that the image sharpness was equal for the bands from the visible part of the spectrum, and approached Nyquist’s theory of digitalization. In the near-infrared part of the spectrum, the MTF values showed a less sharp decrease in comparison with the visible part. Preliminary image acquisition indicates that this hyperspectral system has potential in agronomic applications.

## 1. Introduction

Typical commercial spectrometers or spectrophotometers are usually able to measure the optical spectrum of a specified surface area from a stationary point [1,2,3]. This is done either with one detector scanning the spectrum in narrow wavelength bands, or with an array detector, in which case all the spectral components are acquired at once [4,5]. If the spectrum at more locations is to be measured, the observed object should be mechanically moved in relation to the stationary measuring instrument [6], or vice versa. Therefore, a hyperspectral cube can be created by linear pushbroom sensors in two ways. By moving the hyperspectral scanner, rotation or linear shifts occur on the platform. Surveying using rotation is done by fixing the scanner on a tripod and rotating it so that the axis of rotation and slit line are vertical, and scanning continues in a horizontal direction [6]. Since the hyperspectral images are then created while the sensor line is turning, the geometry of the image is best described as a projection on the surface of a cylinder [7], based on the panoramic camera model [8]. Stands and tripods for laboratory imaging of small objects or calibration of sensors are usually used as moving platforms [9,10]. The potential for using similar hyperspectral sensors is huge, and examples can be found in geological analyses [11], analyses of ceramics and other archaeological artefacts [12,13,14,15,16], deep spectral analyses of works of art [17,18,19], environmental protection [20,21,22], precise agriculture [23,24], the pharmaceutical industry [25,26], medicine [27,28], construction [29] and many other branches of human activity.

Modern CCD cameras, frame grabber boards and modular optical components have made it possible to assemble hyperspectral imaging system with off-the-shelf electronics. Hyvarinen et al. [30], Mao [31], Inoue and Penuelas [32], Yang et al. [33] and Ye et al. [34] developed a hyperspectral imaging device by combining a prism-grating-prism (PGP) spectrometer with a broad range monochrome matrix camera. The integration of a PGP spectrometer and focal plane scanner is an approach to laboratory and field pushbroom hyperspectral imaging [31]. The hyperspectral focal plane scanner scans an input image within the focal plane of a front lens to accomplish the task of pushbroom scanning, while the PGP spectrometer disperses an input line image vertically, as a function of the spectral wavelengths, and focuses a two dimensional output spectral matrix onto the sensor surface of a CCD camera. This integration eliminates the need for a mobile platform for ground and laboratory data acquisition and allows the imaging system to be used for both airborne, field and laboratory applications [33]. Other types of hyperspectral systems are described in [35] (a rotating-mirror scanning hyperspectral imaging device) and [36] (transition from single pixel spectrometers to an FPA base imaging spectrometer). These approaches are implemented by employing a rotating filter wheel or electrically scanned optical filter in front of a monochrome area camera. 2-D imaging of one wavelength at a time is suited to laboratory applications with a stationary sample. However, the previously described approaches are the right solutions in both cases for a moving target (or instrument), a target varying quickly in time, and for laboratory applications with a stationary sample [30].

In this paper, the design of a low-cost hyperspectral system for laboratory and field (terrestrial) data acquisition, which collects sequential continuous samples to create a 3-D image (hyperspectral cube) and spatial distortions assessments is presented. It is one result of research conducted within four international and domestic scientific projects: Airborne Minefield Area Reduction (ARC) [37] Space and airborne mined area reduction tools (SMART) [38], System for Multi-sensor Airborne Reconnaissance and Surveillance in Crisis Situations and Environmental Protection [39] and Toolbox implementation for removal of antipersonnel mines, submunitions and UXO (TIRAMISU) [40] within the domain of humanitarian mine action and crisis situations. Its primary application was the airborne collection of hyperspectral information [41], then creating a 3-D image from continuous samples [42] of suspected hazardous areas, and examining the possibilities of distinguishing vegetation growing over buried mines from vegetation growing outside minefields using spectral reflectance data [43]. Since we are dealing with gathering hyperspectral data on small objects at close distances (up to 3 m), the geometric reconstruction needs to be accurate in order to detect small differences in the spectrum for objects of interest. The low-cost laboratory and field hyperspectral imaging system introduced is similar to systems described in [30,31,32,33,34], but the novelty in this paper is the presentation of the method and results of spatial distortion assessments. Hence, calibration of the developed linear hyperspectral system was carried out. Calibration of linear sensors is usually done by simulating an array sensor [44], and the method described and presented in [45,46] was applied in this work. The modulation transfers function (MTF) [47] is the most frequently used scientific method for describing the performance of optical systems. There are many solutions for calculating MTF, but they are based on determining the parameters for 2-D images [48,49,50]. Due to the specifics of the linear scanner and aim of determining imaging quality at certain wavelengths, spatio-spectral MTF was performed in this work. Finally, the examples of vineyards in the field environment are presented. These data can be forwarded to various experts to provide them with useful insights and new information.

The paper is structured as follows: After this Introduction, Section 2 describes an imaging spectrometer instrument design, its characteristics and specifications. Methods for creating a hyperspectral cube and system calibration are also described in this section. Section 3 provides insights into the results of spatial distortion assessments and the system itself (hyperspectral cubes). Finally, an unconventional method of determining spatial distortion and its results, along with the results of the modulation transfer function of the pushbroom scanning system, are discussed in the last section.

## 2. Materials and Methods

### 2.1. Hyperspectral Imaging System Design

The ImSpector V9 hyperspectral line scanner (HSLS) was combined with a PixelFly monochrome area camera to form the Imaging Spectrograph System, or geometric sensor model (Figure 1).

The ImSpector V9 (Figure 1a) is an imaging spectrometer instrument based on an imaging spectrograph that is “capable of simultaneously measuring the optical spectrum components and the spatial location of an object surface” [44]. The ImSpector V9 employs a direct sight (on-axis) optical configuration and a volume-type holographic transmission grating. This grating is used in a patented prism–grating–prism construction (PGP element, Figure 2), which provides a high diffraction efficiency and good spectral linearity [51]. 

It is almost free of geometrical aberrations because of its on-axis operation, and it is independent of incoming light polarization because only transmission optics were used [51]. The robust structure of the ImSpector V9 suits both industrial and scientific applications that require rapid, precise spectral measurements at a low cost. The spectral resolution of the spectrograph depends on the width of the entrance slit and the linear dispersion produced by the spectrograph optics. The minimum limit for the spectral resolution is set by the imaging capability of the optics. The configuration we used consisted of an optical system with a focal length of 23 mm (Schneider Kreuznach Xenoplan 1.4/23) and a narrow slit 8.8 mm long and 50 μm wide (Table 1), which enabled a spectral resolution of 4.4 nm for 95 channels in a spectral range from 430 nm to 900 nm (Figure 1b, Table 1). At the nadir, the system provides imaging of a narrow strip measuring (0.333 × h) × (0.00208 × h), where h denotes the height above the sample (or ground) (Figure 1b).

PixelFly is a high-performance digital 12-bit CCD area (matrix) camera with a scan area of 8.8 × 6.6 mm and effective pixels of 1280 (H) × 1024 (V) (Scientific type, Table 1). It consists of an ultra-compact camera head, which either connects to a standard PCI or a compact PCI board via a high-speed serial data link. The available exposure times range from 5 μs to 65 s [52]. This particular camera was chosen because the CCD sensor records the light information for each pixel as a grey level with 12-bit dynamic range. Although camera control software enables the 12-bit data of each pixel to be converted to an 8-bit triplet (P_RED_, P_GREEN_, P_BLUE_) [53], this capability was not used to produce the hyperspectral images. After the assembly procedure, the PixelFly camera and ImSpector V9 must be aligned so that the spatial axis of the spectrograph is parallel to the horizontal pixel lines of the camera.

The sun shines at elevation angle *ϴ* measured from the horizontal axis. Waves reflected from the ground surface are collected at the nadir by an optical lens and receiving pattern diagram *F*(*ϴ*). Incident waves are collected by a diffuse collector with a receiving pattern diagram *F*(*α*) (Figure 3a). Other characteristics of the camera include its horizontal and vertical binning capability. Horizontal binning determines the cross-track width of the image, while vertical binning determines the number of spectral bands. The horizontal axis (1280 or 640 pixels) shows the spatial axis and calibrating part, while the vertical axis (1024 or 512 pixels) shows wavelengths from 430 nm to 900 nm (Figure 3a). The dimensions of the analyzed area below the HSLS in static mode (no movement) were defined (Figure 3b). The HSLS output was 1280 × 1024 pixels (or 640 × 512 pixels), whereas 1170 × 1024 or 585 × 512 pixels represent 12-bit intensity of the reflected waves, and 110 × 512 or 55 × 512 pixels on the right side of the image contain information about incident waves. The horizontal x axis is a spatial axis with width *W*, and it contains 1170 or 585 pixels. For the acquired area, the HSLS V9 and PixelFly digital area camera produced from 12.5 (binning factor combinations: H1-V1 and H1-V2) to 24 (binning factor combinations: H2-V1 and H2-V2) images per second, each with 1280 × 1024 pixels (factors 1) or 640 × 512 pixels (factors 2). The vertical axis *λ* shows wavelengths in a range from 430 to 900 nm. The ratio of reflected *E_ref_* and incident *E_inc_* values at wavelength *λ_i_* is the measure of the reflectivity at this wavelength (Figure 3b). The HSLS V9 and image parameters are managed using *Recorder* software, written entirely in C++ language as part of the ARC project. Recorder is an application that actually controls the HSLS V9 (exposure time and horizontal binning factors) and performs image acquisition.

The values *E_inc_* and *E_ref_* measured by the HSLS V9 enable calculation of the reflection coefficient *r* according to the relation:(1)$$r=\frac{\frac{G_{ref}-E_{ref}}{F\left(\alpha \right)}}{\frac{G_{inc}-E_{inc}}{F\left(\theta \right)}}$$
where the pattern diagram of the HSLS V9 is *F*(*ϴ*), and that of the diffuse collector is *F*(*α*); the gains (if other than 1) *G_inc_*, *G_ref_* are of the bands measuring *E_inc_* and *E_ref_*. Thus, we did not measure radiance but calculated the reflection coefficient according to the actual measurements of the reflected and incident portions of the electromagnetic spectrum. The reflection coefficient is the ratio of the amount of electromagnetic radiation recorded by the sensor (*E_ref_*) to the amount of electromagnetic radiation recorded by the diffuse collector near the sensor (*E_inc_*). It is a property of the observed material and is equivalent under different illumination conditions.

Since this hyperspectral imaging system was designed for airborne remote sensing applications, it also consists of a subsystem for navigation, position determination, and orientation of the system in space. This subsystem consists of a single-frequency GPS device integrated with an inertial measurement unit (IMU) iVRU-RSSC by iMAR GmbH with a working rate of 200 Hz, and additional GPS units arranged in or on the platform. IVRU-RSSC is a triple-axis inertial system with three mutually perpendicular MIL-MEMS gyroscopes used to determine the angle elements of the spatial orientation of the sensor and three MEMS accelerometers used to determine the acceleration components along all three axes. Its characteristics are: gyro performance resolution of <0.001°/s; attitude/heading resolution of <0.01°; attitude accuracy of <1° roll/pitch (static or linear unaccelerated motion) and <1° roll/pitch with velocity aiding; digital resolution of 18 bit; and bandwidth of 0–50 Hz (www.cbil.co.uk/products/imar/ivru-rssc/). The bandwidth used in the airborne hyperspectral survey was 50 Hz for gyroscopes and accelerometers, to allow four cycles for the strap down processor to calculate the navigation solution. The accuracy of the geocoded hyperspectral lies in the range of ground sampling distances (GSDs) of the hyperspectral scanner, after performing position and attitude calibration. More details about the calibration and accuracy of geocoding can be found in [54]. This six-degrees-of-freedom measurement unit provides accurate and instantaneous positional data at a high sampling rate, which is continuously recorded during surveying and is synchronized with image data (spectral lines). Thus, if it is necessary to define the precise position of certain spectral data in space, the subsystem can be activated to collect the required data to execute parametric georeferencing. Although the subsystem was not used to collect the spectral data presented later in this paper, it has this functionality, and could be applied in possible future applications such as mobile hyperspectral mapping.

### 2.2. Sensor Bracket and Control System

For the purpose of terrestrial imaging within the TIRAMISU project (2012), a sensor bracket and control system were built. An aluminum construction with a 3-metre guide-rail was made (Figure 4), along which the HSLS V9 could move and collect spectral samples continuously. The source of light used during imaging in the laboratory was a halogen lamp, which, due to its smooth spectral properties, had no significant peaks over the entire visible and NIR spectra. To enable movement and manipulation while collecting spectral samples in succession, a control system with an engine and software was designed. Each time the program started (including charging), initialization and alignment of the system were carried out by selecting *Init*. This sub-program issues the command to place the system at the zero point on axis X, which is the starting point (far left margin) on the guide (from the perspective of the operator). The maximum distance from zero to the farthest point is x = 2,450 mm, and the speed ranges from 1 to 200 mm/s. The software enables the following recording regimes:
Continual movement and collection of spectral samples at a given speed to the given position.Movement at the given speed by sections of the route, stopping at positions where spectral samples are collected.

### 2.3. Spatial Calibration and Modulation Transfer Function

Hyperspectral sensors used in close-range imaging applications are primarily calibrated in the spectral domain, while their spatial calibration is questionable or not implemented at all. To (geo)locate the spectral information gathered by the HSLS V9 sensor, a mathematical model of the imaging of the scene to the image plane was derived and implemented, taking into consideration the expected systematic errors that influence measurement results. A detailed description of the applied 2-D method can be found in [46]. In brief, it features an algorithm that rests on the theory of the central projection of each point in space in the image plane, using collinearity Equations (2). This mathematical model is the standard one used when calibrating cameras with array sensors for photogrammetric purposes. The observation equations for the *j*-point on the *i*-th image are given as follows [38]:(2)$$\begin{array}{l} \vartheta_{\xi ,i,j}={\left(\frac{\delta \xi }{\delta \xi_{0}}\right)}_{i,j}^{0}d\xi_{0,i}+{\left(\frac{\delta \xi }{\delta c}\right)}_{i,j}^{0}dc_{i}+{\left(\frac{\delta \xi }{\delta \alpha }\right)}_{i,j}^{0}d\alpha_{i}+{\left(\frac{\delta \xi }{\delta x_{0}}\right)}_{i,j}^{0}dx_{0,i}+{\left(\frac{\delta \xi }{\delta y_{0}}\right)}_{i,j}^{0}dy_{0,i}\\ \;\;\;\;\;\;\;\;\;\;\;\;\;\;\;\;\;\;\;+{\left(\frac{\delta \xi }{\delta x_{T}}\right)}_{i,j}^{0}dx_{j}+\;{\left(\frac{\delta \xi }{\delta y_{T}}\right)}_{i,j}^{0}dy_{j}+{\left(\frac{\delta \xi }{\delta K_{1}}\right)}_{i,j}^{0}dK_{1,i}+{\left(\frac{\delta \xi }{\delta K_{2}}\right)}_{i,j}^{0}dK_{2,i}\\ \;\;\;\;\;\;\;\;\;\;\;\;\;\;\;\;\;\;\;+{\left(\frac{\delta \xi }{\delta K_{3}}\right)}_{i,j}^{0}dK_{3,i}-\left(\xi_{i,j}-\xi_{i,j}^{0}\right) \end{array}$$
where:
$\xi_{T}$ is the image coordinate of the measured point.$\xi_{0}$ is the position of the principal point of autocollimation in the image coordinate system.*c* is the sensor’s principal distance.$x_{T},y_{T}$ are the object’s reference coordinates for the measured point.α is the angle of the sensor’s optical axis in the reference coordinate system, and$K_{1},\;K_{2},\;K_{3}$ are the radial distortion coefficients of the 3th, 5th, and 7th orders.

To determine the parameters of the line sensor calibration, i.e., the position of the principal point (ζ_0_), the calibrated focal length (*c*), and the coefficients of the distortions *K*_1_, *K*_2_, and *K*_3_, it is necessary to make at least five independent observations. In this case, the central projection of a point in space was used in a two-dimensional image plane. However, with a line scanner there is only one dimension, so a mathematical model was set up and resolved so that the image coordinate system was reduced to only one dimension [45]. Hyperspectral measurement by the HSLS V9 was performed along the line of the black and white sample with the repetitive crown patterned target surface (Figure 5). When calibrating the line scanner to the given sample, the aim was for the line of imaging to pass along the line joining the center of edge markers A and B, with a black-and-white triangular sample (Figure 5, red line). Focusing was performed on the basis of visual judgment.

MTF is the ratio of output modulation (sinusoid wave) to input modulation (square wave) (4) normalized to unity at zero frequency, while modulation is the variation (*V_max_* and *V_min_*) of a sinusoidal signal about its average value (3) [56]:(3)$$MODULATION=M=\;\frac{V_{max}-V_{min}}{V_{max}+V_{min}}$$
(4)$$MTF\left(f\right)=\;\frac{M_{OUTPUT}\left(f\right)}{M_{INPUT}\left(f\right)}$$

MTF is the magnitude response of the optical system to sinusoids of different spatial frequencies, and it is a measure of how well a system will reproduce the input object faithfully. An object or image-plane irradiance distribution is composed of spatial frequencies in the same way that a time-domain electrical signal is composed of various frequencies—by means of a Fourier analysis. By taking a one-dimensional profile across a two-dimensional irradiance distribution, we can obtain an irradiance vs position waveform, which can be Fourier decomposed in exactly the same way as if the waveform was in the more familiar form of volts’ vs. time [47]. There are two general methods of determining MTF. The first is the direct method based on measuring the response of the sinusoidal signals of the decreasing bar widths on the recorded template. The second is an indirect method based on calculating Fourier’s transformation of the linear transference function. We used the direct method approach as a handy reality check, comparing a measured spot size to calculated MTF values. The target for measuring and calculating MTF was the binary pattern of black and white pairs of decreasing bar widths that simulated different object sizes, as seen in Figure 6a, which was easy to make [56]. The increase factor of the black and white bars was 2^1/6^ [57], which means the width of pairs was calculated as between 20.88 mm (widest) and 0.92 mm (narrowest).

The calculation of MTF at any particular frequency requires that a CTF calculation (5) is made at a series of frequencies harmonically related to the desired frequency of the MTF measurement:(5)$$CTF=\;\frac{I_{max}-I_{min}}{I_{max}+I_{min}}$$
where *I* is the average intensity of black (*I_min_*) and white (*I_max_*) bars. For an infinite square wave, CTF is defined as the image modulation depth as a function of spatial frequency. Series conversion between CTF (6) and MTF (7) using Fourier decomposition of the square waves can be derived as follows [40]:(6)$$CTF\left(f_{x}\right)=\frac{\pi }{4}\times \left|MTF\left(f_{x}\right)+\frac{MTF\left(3\times f_{x}\right)}{3}-\frac{MTF\left(5\times f_{x}\right)}{5}+\frac{MTF\left(7\times f_{x}\right)}{7}+\frac{MTF\left(11\times f_{x}\right)}{11}\right|$$
And:(7)$$MTF\left(f_{x}\right)=\frac{\pi }{4}\times \left|CTF\left(f_{x}\right)+\frac{CTF\left(3\times f_{x}\right)}{3}-\frac{CTF\left(5\times f_{x}\right)}{5}+\frac{CTF\left(7\times f_{x}\right)}{7}+\frac{CTF\left(11\times f_{x}\right)}{11}\right|$$

This operation must be repeated for each frequency at which we want to find MTF. Typically, CTF needs to be measured at enough frequencies to plot a smooth curve and then interpolated to find the CTFs at the frequencies needed to compute an MTF curve from the CTF data. It is generally not sufficiently accurate to take only the CTF measurements as MTF measurements (Figure 6b), but this can be a good estimate of MTF behavior [40].

### 2.4. Creating a Hyperspectral Cube

The spatial accuracy of continuous measurements depends on the movements of the system. When the system is on a moving platform, the ground can be scanned linearly at line intervals Δs and GSD_v_ along the line. Line interval Δs depends on the velocity of the system (v) and frequency of storing images (f_s_) in the acquisition system. To use the line scanner in full imaging mode (acquiring contiguous scan lines), it is necessary to find the optimum speed of the HSLS V9. It is a function of the required GSD and scanner imaging frequency, according to a simple Equation (8), used to arrive at optimum distance per second (according to [42]):(8)$$S=\;\frac{\mathrm{GSD}}{f_{i}}\;\;$$
where:*S* is the speed of the HSLS V9 (m/s).GSD is the Ground Sampling Distance across the line scanner (m).*f_i_* is the imaging scan period (s).

The maximum frame rate per second was around 24 Hz and depended on radiometric parameters during the collection of reflected radiation. The HSLS V9 requires a very complex calibration procedure and time-consuming processing. Software solutions were also developed to produce a raw hyperspectral cube (merging a large number of continuous line samples in a two-dimensional image without georeferencing) in the Matlab software package. These operations were automated, with code written in the Matlab development environment specifically for this purpose, and contained the following procedures:Exporting raw data (linear images) and transforming them into TIF format in Recorder program.Creating mean data on insolation collected with the diffuse collector, along each line.Calculating the reflectivity coefficient.Correcting the spectral responses with dark current data (dark image subtraction).Stacking spectral lines in the hyperspectral cube.

When creating the hyperspectral cube, we also took into account the appearance of bright pixels at long exposure times due to sensor lattice damage (hot pixels), which can be corrected via dark image subtraction. Hot pixels in the sensor must be corrected by characterizing their behavior as a function of exposure time when they are not exposed to any light. This can be achieved by closing the aperture completely and obtaining several measurements. By obtaining the baseline pixel values for several exposure times, hot pixel background signals can be modelled in each pixel by [35]:(9)$$h_{bg,ij}=a_{ij}\times T_{exp}+b_{ij}$$
where indices *i* and *j* indicate the pixel’s position in the sensor, *a_ij_* and b_ij_ are the coefficients that model the linear behavior of the pixel as a function of exposure time *T_exp_*, and *h_bg,ij_* is the estimated hot pixel value that should be subtracted from each pixel for every new measurement. The data are finally converted to create a pyramid in BSQ formation with an ENVI header. Thus, it is possible to download the raw hyperspectral cube directly in PARGE 2.3 software, in which parametric georeferencing [58] of the hyperspectral cube is carried out.

The procedure for parametric georeferencing involved adding the exact spatial position of each pixel in the hyperspectral cube. This procedure required, in advance, the given elements of external orientation (GPS and IMU data) for each line in the cube. During data collection, the *Recorder* software along with spectral lines created a table in which the GPS and IMU data were linked with each stored line. Thus, the fine synchronization of images and metadata necessary for the parametric georeferencing of the hyperspectral cube could be performed. This procedure was also fully automated by applying the script in C# programming language. Only data recorded during the scanner operation at its frequency were used, while the rest of the considerable data from iMAR were rejected as unnecessary, in order not to overload the computer resources. Using this program, apart from interpolation, the data can be acquired in a format adapted to the PARGE parametric geocoding program. Due to the mutually independent data groups (linear images with the HSLS, their metadata, and data from IMU), and various programs used to access them (MatLab and C#), this configuration allows parallel processing on multicore systems, which speeds up the overall data processing chain considerably. A detailed description and explanation of the parameter georeferencing process can be found in [59].

## 3. Results

Once all the characteristics, properties, and constraints of HSLS V9 and the presented program codes were defined, the imaging system needed to be characterized and verified to see that it was behaving as described. The images were obtained from the HSLS V9 (explained in Section 2.1 and Section 2.2). The focus was on the overall results of the reviewed system introduced in this article with all its elements: the hyperspectral imaging system design (HSLS V9), sensor bracket, control system, program codes for creating the hyperspectral cube, spatial calibration, and MTF calculation of the hyperspectral line scanner. In this section, we provide the results of MTF calculations, spatial calibration results in laboratory conditions, and an example of hyperspectral surveys in the field, to demonstrate the use of the HSLS V9 as presented.

### 3.1. Spatial Calibration Results and Calculation of the Modulation Transfer Function

Before the survey mission, it was necessary to examine the imaging process of the linear scanner in order to perform spatial calibration. Three images of the different camera orientations in relation to the calibration pattern were taken, so that the homologous rays of photogrammetric bundles were intersected as closely as possible to right angles (90 degrees). This ensured the best geometric conditions for the intersections of the homologous rays and, thus, the precise determination of the elements of the inner orientation (Table 2). Imaging was performed at a distance of 3 m, and the calibration was performed separately by spectral channels. The results of the distortion values of some wavelengths, changes in the camera constant (linear scanner), and shifts of the principal point (central pixel) are shown in Figure 7. The diagrams in Figure 8a–d show the distortion values on the spectral line of the central pixel (zero value) to the margin pixel of the scanner line (450 pixels value). The diagrams in Figure 8e,f show shifts in the camera constant and principal point of the HSLS V9 system.

The value of the distortion curves shows the distortion values within a range of 0 to 11 for the visible part of the spectrum (Figure 7a,b), while distortion rises drastically at the edges of the image for the near-infrared portion, above 25. This is the consequence of the lack of correction of the lens for chromatic dispersion for the infrared portion of the spectrum. The analysis of the derived results further shows that as the wavelength changes, the camera constant and position of the principal point also change significantly (Figure 7e,f). The position of the principal point changes almost linearly, within a range of 499.9 pixels for the blue-indigo portion of the spectrum to 486.8 pixels for the infrared portion, while the camera constant changes from 3614 pixels for the blue-indigo portion of the spectrum to 3539 pixels for the near-infrared portion (Table 2). This change is of a linear character (Figure 7f).

The mathematical calibration model defined in [38,39] and used to determine distortions reflects the physical reality of the imaging procedure and is analogous to a photogrammetric solution of the central projection of the area in the image plane, degraded to only one dimension (a line). Only radial distortion was considered, since tangential distortion in line sensors has no influence. To this end, the Brown distortion model with seventh-degree polynomial correction was implemented.

Imaging of the sample in order to determine MTF for the HSLS V9 was performed at a distance of 1.7 m, with the camera placed so that the slit on the lens of the Imspector V9 was at right angles to the line of the samples. This distance was selected because imaging is performed at a similar distance in laboratory and field condition, depending on the height of the construction and samples. The result of the imaging for the purpose of determining MTF was a 2-D image, on which the x axis measures the reflectivity of spectral data of the recorded scene for individual wavelengths, and the y axis shows the wavelengths on which the spectral values of each linear sample were recorded (Figure 8a). During imaging, these calibration parameters were used to reduce the distortion effect. Since the spectral component of the adapted HSLS V9 was more important during imaging, the motivation for this research was to investigate and detect changes in the geometric quality linked to certain wavelengths, rather than attempting to determine the geometric quality of the image without errors. MTF was calculated for 10 spectral lines along the entire spectrum from 430 to 900 nm on the image by the PixelFly sensor, starting from 0 (the top edge of the image) at intervals of 100 pixels (Figure 8b), in order to establish MTF changes, depending on the wavelengths, under the same conditions, and with the same focus.

Theoretical spatial resolution along the line sensor (GSD_v_) was 0.48 mm for binning factor 1 ((1.7 m × 0.333)/1170) or 0.97 mm for binning factor 2 ((1.7 m × 0.333)/585). The values of actual (real) GSD_h_ determined on the spectral profile lines were from 100 to 1000 pixels. The values of actual GSD_v_ for 4 spectral profiles for 100, 500, 700, and 1000 pixels are listed in Table 3. The derived MTF results (Figure 9) indicate that the sharpness of the images throughout the visible part of the spectrum was equal, and approached Nyquist’s theory of digitalization, which should be the result of a specially constructed, high-quality lens for hyperspectral measurement (Schneider Kreuznach Xenoplan 1.4/23). This mostly depends on the resolution of the CCD sensors (in this case, the PixelFly array camera). In fact, the MTF coefficient value fell continuously from the starting value of around 0.7 to 0.1, while the coefficient value in the near-infrared part of the spectrum showed a milder fall. The relationships between theoretical and actual determination along the line sensors (GSD_v_) is shown in Table 3. An analysis of spatial resolution along the line scanner GSD_h_ was not performed.

### 3.2. Field Hyperspectral Surveying of Vineyards

In 2014, a trial was carried out in Jadrtovac, Croatia, to simulate various conditions of providing vines with water, consisting of three irrigation variants and a control variant without irrigation. The irrigation variants differed in regard to the amount of water provided in relation to the vines’ requirements for water. These requirements were defined by taking into account the average referential evapotranspiration (ET) data (30-year average) from the Šibenik meteorological station, precipitation at the trial plantation, and the features of the vines in the area [60]. The variants and trial labels were:Variant A: 50% ET—irrigation providing 50% of the calculated requirements of the vines for water (delivered by a single pipe with a diameter of 1 cm).Variant B: 75% ET—irrigation providing 75% of the calculated requirements of the vines for water (delivered by two pipes with diameters of 1 cm).Variant C: 100% ET—irrigation providing 100% of the calculated requirements of the vines for water (delivered by three pipes with diameters of 1 cm).Control variant: no irrigation.

The hyperspectral pilot survey of the plantation in the location was carried out on 30 July 2015. All four rows were surveyed (the control and three irrigation variants) in two sections, each 2.4 m long. Dark current imaging was used. The HSLS V9 was used with binning factor 2 and a default gain value of 1, between 12:30 and 14:45, from a metal construction fixed to a trailer pulled from site to site by a tractor (Figure 10a). The actual surveying was performed when the construction was at rest and its engine was operating the HSLS V9. The stony surfaces between the rows were covered in dark plastic netting in order to reduce the effect of light from the surroundings. Binning factor 2 was used because the area between the rows of vines was homogeneous, and the data on spectral response from the area for further processing were more important, in other words, calculating 16 different types of vegetation indices.

For the same reason, parametric georeferencing of the hyperspectral cubes was not performed. The system was placed approximately 3.7 m above the ground, while the tops of the vines were about 1.2 m from the ground. Thus, the tops of the vines (top leaves) were about 2.5 m from the lens. Accordingly, the theoretical spatial resolution (GSD_v_) along the line sensor with binning factor 2 was 1.4 mm, and the theoretical spatial resolution (GSD_h_) across the line sensor was 5.2 mm. GSD_v_ ≈ 1.4 mm, the frame rate was 24 fps, the exposition time was 0.042 s, and the speed of the HSLS V9 was calculated according to (8) at 33 mm/s. Raw hyperspectral cubes were created from the spectral lines, as described in Section 2.3 (without parametric georeferencing) for each part of the row surveyed. Then, all four cubes were merged in a mosaic showing all four rows next to each other (Figure 10b). Quick atmospheric corrections in the ENVI program package were carried out on the mosaics of all four hyperspectral cubes.

After processing the hyperspectral cubes, the ENVI program package was used to calculate 16 selected vegetation indices, which served agronomists as input data for the complex analysis of chlorophyll and moisture in the vines [60].

## 4. Discussion and Conclusions

This article focuses on defining and describing the HSLS V9 low-cost system for collecting hyperspectral data (cubes) and spatial distortion assessments. The HSLS V9 line scanner based hyperspectral imaging system described in this paper shows promise for agricultural (grapevine) applications. It may be a useful remote sensing tool for hyperspectral remote sensing methods in laboratory and terrestrial applications. The horizontal and vertical binning capability of the line scanner allows images with various spatial and spectral resolutions along the line scanner (GSD_v_) to be created. The intention behind the use of the HSLS V9 is to survey a scene or samples from a construction at a distance of about 3 m, and the emphasis is on the spectral and spatial characteristics of the recorded objects. Therefore, developing techniques for geometric correction was not the main objective of this paper. But a six-degrees-of-freedom measurement unit (IMU) that provides accurate and instantaneous views and positional data with a high sampling rate is part of the HSLS V9, and the resulting data can be used to correct geometric distortions for a specific task if necessary.

The most common approach to calibrating the line sensor in close-range applications is to derive the calibration model using one 3-D-pattern image specifically constructed for the particular application [61]. Another concept is to extend one-dimensional imaging by one more spatial dimension [62]. The mathematical model used in this research of linear array sensor imaging is transformed to the mathematical model (with one line and many columns), for which many calibration algorithms have already been developed. Thus, the mathematical model of the spatial calibration of the line scanner differs from the mathematical model applied in spatial photogrammetric reconstruction. In this model, the mathematical model of calibration corresponds to the mathematical model of spatial reconstruction. This enables the impact of calibration results on the quality of spatial reconstruction to be tracked and analyzed. So, using the parameters of inner orientation obtained by the proposed calibration method in the geocoding of hyperspectral cubes, a significant increase in the positional accuracy of every geocoded pixel in the hyperspectral cube can be achieved. In fact, this means producing a mathematical imaging model aligned with the ideal central projection of the area in a 1-D sensor model, extended to account for the distortion influence using the Brown distortion model with seventh-degree polynomial correction.

Compared to the usual calibration model of industrial line scanners, it should be noted that this model is adapted to industrial applications, where quick, automatized calibration on a pre-defined sample must be carried out, and during which the laws of central projection of a line from the area to a line on the image and determination of equations of collinearity are used for the mathematical model. This avoids the limiting requirement (for conducting photogrammetric calibration) that the sample must be in a single plane. This model is used to determine the effect of radial distortion.

The results of the research show that the inner calibration parameters depend significantly on the wavelength of the light falling on the sensor. Similarly, the degree of this influence depends crucially on the quality of the lenses used in the multispectral or hyperspectral cameras. The use of lenses that are carefully corrected for chromatic dispersion (achromatic, or better still, apochromatic) is recommended, as they reduce the effect of light wavelengths on calibration elements to some extent, although they do not rule it out entirely. In this, the trend of change in the position of the main point is almost completely linear, ranging from 499.9 px for the blue-indigo part to 486.8 px for the infrared part of the spectrum, with a mean determination error of around 0.1 px. The focal distance also changes from 3614 px (blue-indigo channel) to 3539 (near infra-red channel), and the change is non-linear. These results clearly show that the light wavelength used when calibrating the hyperspectral scanner must be taken into account.

For high-quality modelling of the calibration parameters, calibration must be performed for the border and median wavelengths of the spectrum within which imaging is to be carried out. In exceptional cases, calibration can be performed using the middle part of the spectrum. The effect of distortion is not negligible either, and ranges from 4 px for the blue-indigo to 25 px for the near infra-red part of the spectrum (on the line edges), where the distortion effect increases sharply. Therefore, when calibration is performed, it is absolutely necessary to account for distortion parameters and use them to eliminate systematic distortion in the imaging results.

An assessment of the quality of HSLS optical imaging, using the modulation transfer function, has shown that the sharpness of the imaging throughout the entire visible part of the spectrum is the same and approaches Nyquist’s digitalization theory. This is the consequence of a specially constructed, high-quality lens for hyperspectral surveying, and depends mainly on the resolution of the CCD Pixelfly sensor. In fact, the coefficient of the modulation transfer function falls almost linearly from an initial value of 0.7 to 0.1, with a borderline frequency of around 0.3 cy/px for blue, and to 0.6 cy/px for near infrared. It is interesting to note that the borderline frequency for the near-infrared area is almost twice as high in relation to the visible part of the spectrum.

The results presented in this article in the form of hyperspectral cubes are useful in a spectral analysis of the vines in the Jadrtovac area. The differences in the values of the calculated vegetation indices provides agronomic experts with high-quality input data for a decision support system to manage vines and vine products. However, more research is needed to evaluate the radiometry aspects of HSLS V9 for stationary remote sensing methods because of the restrictions and problems that occurred during research and implementation. Restrictions and problems in gathering hyperspectral data and creating hyperspectral cubes were seen in the poor radiometric quality of some lines. During surveying, sudden changes in the radiometric characteristics of certain lines occurred as a result of exceptionally bright objects on the scene. The next research steps should be to analyze this phenomenon and improve image quality. Further research should move in the direction of finding methods of surveying and processing with the aim of producing hyperspectral cubes with a GSD_v_ spatial resolution instead of GSD_h_.

## Figures and Tables

**Figure 1 sensors-19-04267-f001:**
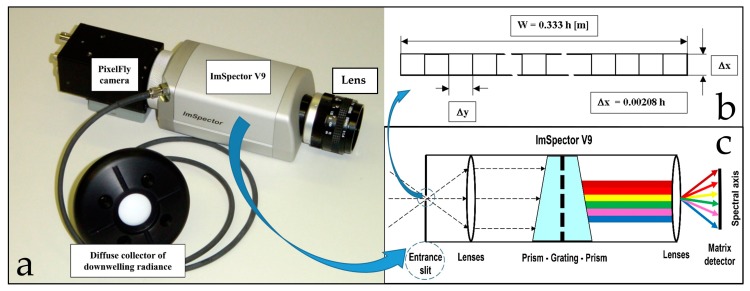
(**a**) ImSpector V9 hyperspectral line scanner with diffuse collector to measure incident down-welling irradiance (connected to the V9 head via fiber-optic cable) and PixelFly matrix camera. (**b**) ImSpector V9 sensor geometry mapping scheme. (**c**) Operating principle of ImSpector V9.

**Figure 2 sensors-19-04267-f002:**
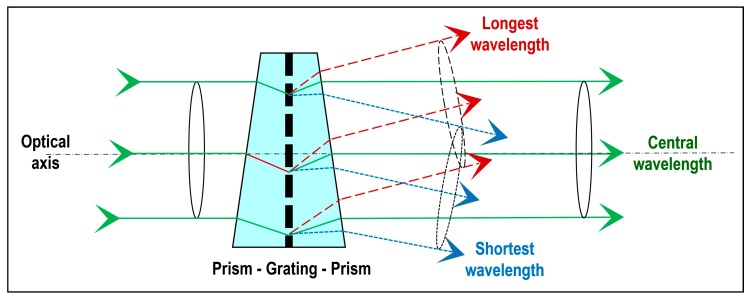
Basic principle of the direct-vision PGP element.

**Figure 3 sensors-19-04267-f003:**
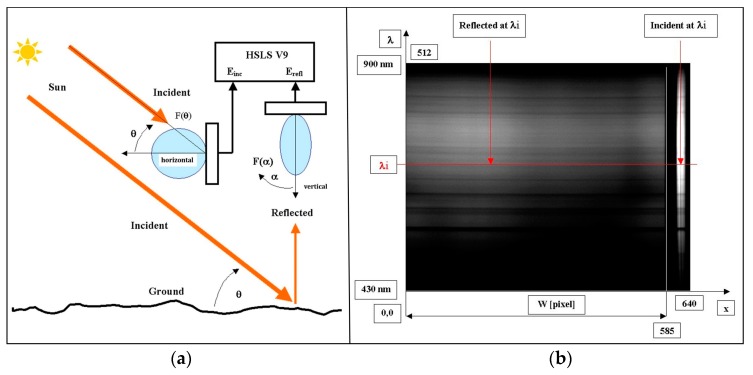
(**a**) Geometry in the vertical plane of measurement with the HSLS V9 (picture courtesy of M. Bajić). (**b**) Output images from the HSLS V9 and PixelFly digital area camera in a spectral range from 430 to 900 nm. Left: 585 × 512 (reflected) pixels show 8-bit intensity of reflected waves. Right: 55 × 512 (incident) pixels at the right side of the image contain information on incident waves in binning II configuration.

**Figure 4 sensors-19-04267-f004:**
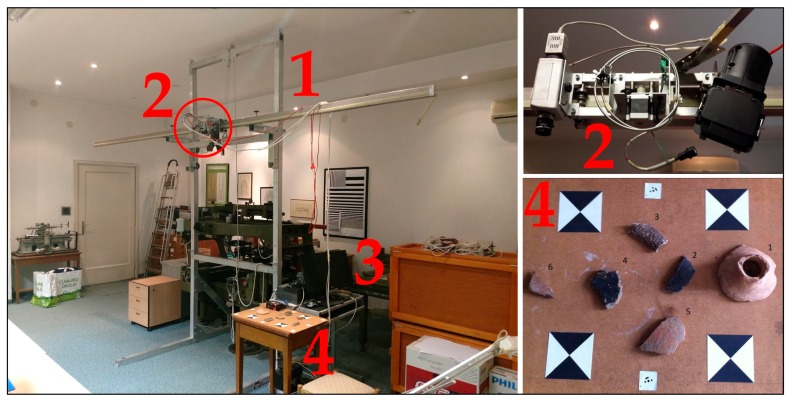
(**1**) Sensor bracket; (**2**) HSLS V9, diffuse collector, halogen lamp, engine (in red circle), (**3**) Control system (industrial controller and motor controller) and (**4**) Table with Roman archaeological artefacts found in Sisak, Croatia [55].

**Figure 5 sensors-19-04267-f005:**
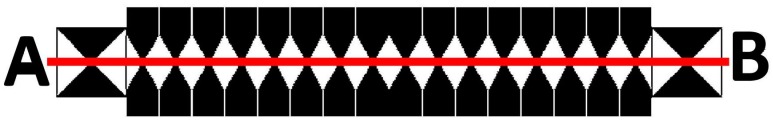
Calibration pattern for spectral calibration of HSLS V9.

**Figure 6 sensors-19-04267-f006:**
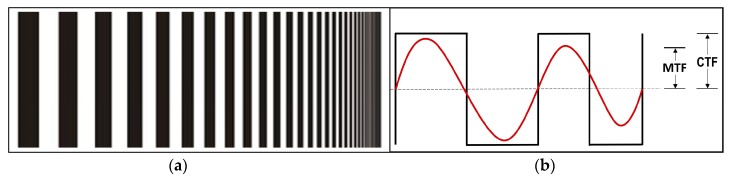
(**a**) Template with striped black and white bars of decreasing width. (**b**) Components of CTF and MTF. CTF is usually equal to or greater than MTF.

**Figure 7 sensors-19-04267-f007:**
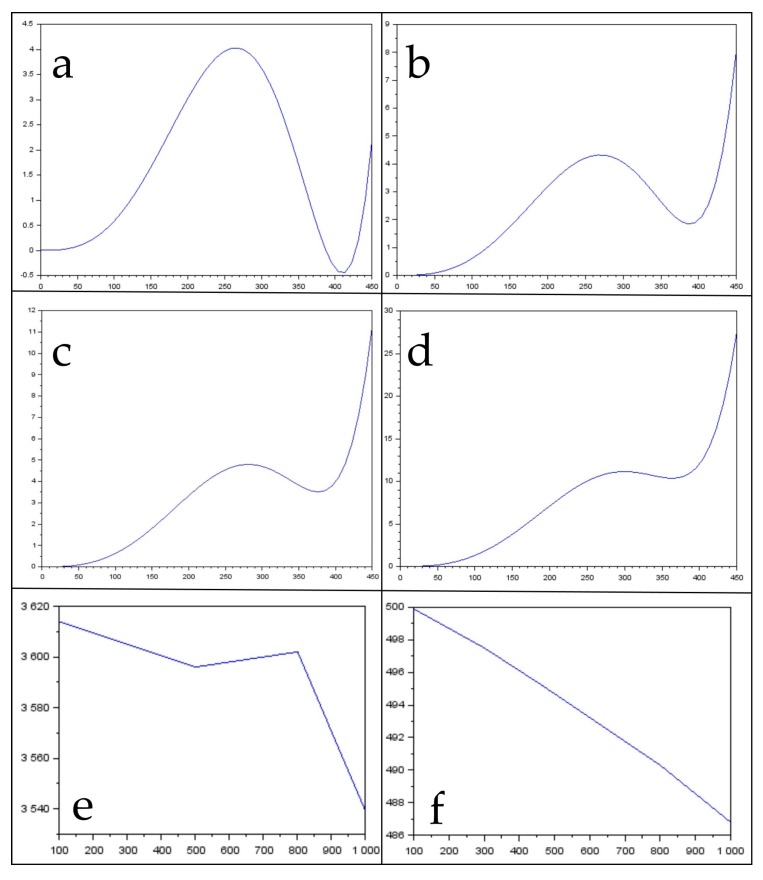
Distortion values for the (**a**) blue-indigo (100 pixels from the top of the image), (**b**) yellow-green (300 pixels from the top of the image), (**c**) orange-red (500 pixels from the top of the image), and (**d**) near-infrared (1000 pixels from the top of the image) parts of the spectrum, while (**e**) shows the camera constant change in relation to wavelength and (**f**) shows the change in the position of the principal point in relation to the wavelength. All values on the axis are expressed in pixels.

**Figure 8 sensors-19-04267-f008:**
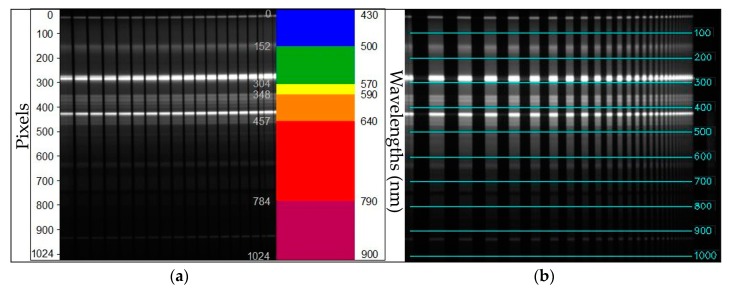
(**a**) Distribution of wavelengths in the raw HSLS V9 image along axis y. (**b**) Positions of the spectral lines for which MTF was calculated (blue transverse lines, while the numbers indicate the ordinal number of the line in relation to the initial line, in pixels).

**Figure 9 sensors-19-04267-f009:**
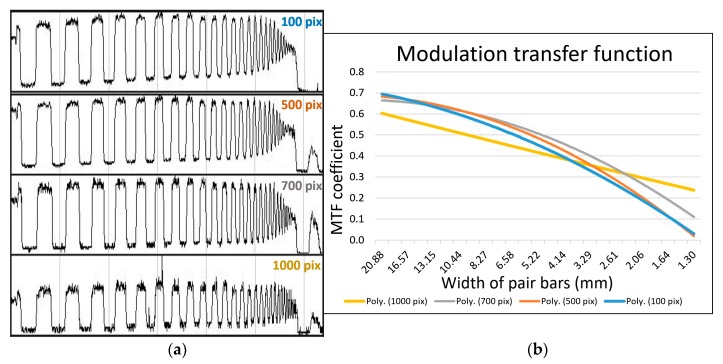
(**a**) Spatial frequencies of spectral profile lines for 100, 500, 700 and 1000 pixels. (**b**) Calculated MTFs according to these profiles.

**Figure 10 sensors-19-04267-f010:**
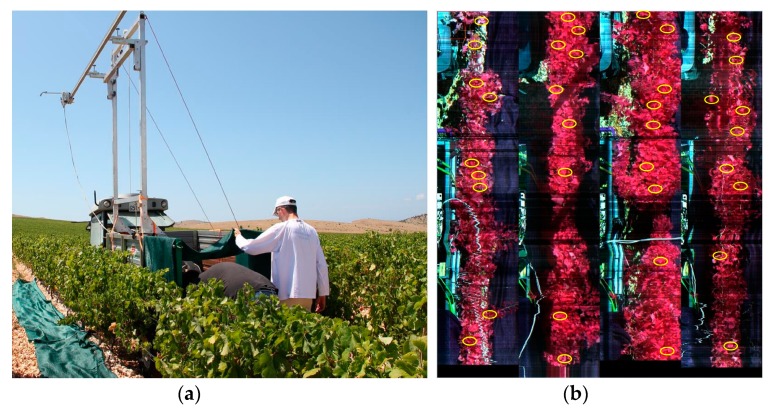
(**a**) Sensor bracket, HSLS V9 and control system during hyperspectral survey in Jadrtovac vineyard. (**b**) Mosaic of four hyperspectral cubes (with a combination of wavelengths: 850, 650, 550 nm) with places marked where samples were taken to calculate vegetation indices.

**Table 1 sensors-19-04267-t001:** Basic specifications of the ImSpector V9 line scanner [44] and PixelFly area CCD camera [45].

**ImSpector V9 Specifications**		
Spectral range	430∓900 nm ± 5 nm	Designed for 6.6 mm detector; corresponding to shorter axis of 2/3” CCD
Spectral resolution	4.4 nm	With 50 µm slit
Numerical aperture	0.18	F/2.8
Slit width	50 µm	
Effective slit length	8.8 mm	
Image size	6.6 mm × 8.8 mm	Corresponding to standard 2/3” CCD
Magnification of spectrograph optics	1x	
**PixelFly Basic Specifications**		
Image resolution	1280 × 1024 pixels	
Pixel size	6.7 µm × 6.7 µm	
Scan area	6.9 mm × 8.6 mm	
Imaging frequency (frame rate)	12.5 fps	At binning with factor 1
	24 fps	At binning with factor 2
Pixel scan rate	20 MHz	
Exposure time	10 µs–10 s	
Binning horizontal: Binning vertical:	factor 1, factor 2 factor 1, factor 2	

**Table 2 sensors-19-04267-t002:** Internal orientation elements for positions of the spectral lines of 100, 300, 500 and 1000 pixels.

*ζ*_0_ (px)	*c* (px)	*K* _1_	*K* _2_	*K* _3_	Positions of the Spectral Lines (px)
499.9 ± 0.1	3614 ± 14	0.660	−0.080	0.0024	100
497.5 ± 0.1	3605 ± 14	0.695	−0.085	0.0027	300
494.7 ± 0.1	3569 ± 14	0.710	−0.084	0.0027	500
486.8 ± 0.1	3539 ± 14	1.459	−0.164	0.0053	1000

**Table 3 sensors-19-04267-t003:** Theoretical and actual (determined) spatial resolution along line sensor (GSD_v_) for the spectral line at 100, 500, 700 and 1000 pixels.

Theoretical GSD_h_ (mm) (Binning factor 1)	Determined (Actual) GSD_h_			
	100 Pixels (mm)	500 Pixels (mm)	700 Pixels (mm)	1000 Pixels (mm)
0.48	0.58	0.58	0.51	0.51

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
