# Peer review of "Spatial Distortion Assessments of a Low-Cost Laboratory and Field Hyperspectral Imaging System"

_sensors, 2019, doi:10.3390/s19194267_

Round 1

Reviewer 1 Report

The paper presents an adapted version of an existing aerial hyperspectral imaging system to collect hyperspectral data.  The system consists of an imaging spectrometer system, a high performance CCD camera, a subsystem for navigation, positioning, and orientation in space, and a control system.  The paper mainly focuses on describing the design of the imaging spectrometer and on determining the imaging modality and limitations.  Applications in agronomy and archeology are shown.

The paper is well-written.  However, the introduction is lacking some information regarding the previous related works particularly those concerned with similar systems/sensors.  I was also lost at the beginning because I could not find the road map of the paper in the introduction: usually, it is provided as the last paragraph of the introduction and states what to find in which section. There is a section numbering problem: section 2.3 is repeated (creating a hyperspectral cube,     Spatial calibration and modulation ...).

To improve the paper, the authors may need to clearly emphasize their contribution and clearly indicate what each section of the paper is addressing.  This will definitely help the reader.

Author Response

Response to Reviewer 1

________________________________________________________________________________________________________

Reviewer: 1

Comments to the Author   

The paper is well-written.  However, the introduction is lacking some information regarding the previous related works particularly those concerned with similar systems/sensors.  I was also lost at the beginning because I could not find the road map of the paper in the introduction: usually, it is provided as the last paragraph of the introduction and states what to find in which section. There is a section numbering problem: section 2.3 is repeated (creating a hyperspectral cube, Spatial calibration and modulation ...).

To improve the paper, the authors may need to clearly emphasize their contribution and clearly indicate what each section of the paper is addressing.  This will definitely help the reader.

Authors Comments

Thank you for the detailed reading of the manuscript and positive comments.

We have supplemented the Introduction in the way you suggest (the previous related works with similar systems/sensors; emphasizing our contribution; clearly indication what each section of the paper is addressing).

We also added the road map of the paper at the end of the introduction.  

The section numbering problem no longer exists, we have corrected the section numbering.

We hope that we have improved the quality of the paper with listed additions and enhancements.

Yours sincerely,

Andrija Krtalić

Vanja Miljković

Dubravko Gajski

Ivan Racetin

Reviewer 2 Report

There are two objectives in this paper: 1. A description of a typical hyperspectral line scanner imaging system for spectral sensing applications and 2. To review the imaging capability of the system and to assess its limitations through spatial/spectral calibration procedures. The paper then gives two data sets collected by the system and to indicate its usefulness for many practical applications.

Here are the comments about this paper:

1.       HSI line scanners have been around for over few decades and there are numerous papers have reported about their properties, characteristics and limitations. However, this paper has been focussed on a specific type of instrument (Imspector) and so it is still justified for a publication despite of abundant of information already in the public domain.  

2.       As far as the reviewer is concerned the strength of this paper may be the distortion assessments for this particular type of HSI line scanner, and the other information should be heavily condensed or even removed to improve the clarity of the paper :

a.       The details of the ‘Recorder’ software such as binning/interface gui etc should be all removed;

b.       The presentation of the experimental data collected by the system is really not necessary, or may be just one example is more than enough.

3.        Suggestions for further improvement:

a.       The paper should be focussed more on distortion assessments of HSI line scanners and so the title should reflect this and something like : ’Spectral & spatial distortions assessments in typical hyperspectral imaging line-scanner system’;

b.       Remove table 2;

c.        Remove the binning info in fig 3;

d.       Give more info about the IMU (line 160) particularly its spatial resolution/accuracy;

e.        Remove fig 5;

f.         Expand section 3.1 and to give more details of fig 8 (which is one of the main results of the paper). Also expand discussions on fig 9 & 10 if possible.

g.        About the experimental result it is suggested to use one example instead of two so to keep the paper more focussed on the distortion assessment.

h.       Questions about the vines data set: what is ‘medium value’ in fig 13? What is the implication of this figure?

i.         The discussion and conclusion is far too short: needs substantial expansion!  

Author Response

Response to Reviewer 2

____________________________________________________________________________

Reviewer: 2

Comments to the Author

There are two objectives in this paper: 1. A description of a typical hyperspectral line scanner imaging system for spectral sensing applications and 2. To review the imaging capability of the system and to assess its limitations through spatial/spectral calibration procedures. The paper then gives two data sets collected by the system and to indicate its usefulness for many practical applications.

Here are the comments about this paper:

HSI line scanners have been around for over few decades and there are numerous papers have reported about their properties, characteristics and limitations. However, this paper has been focussed on a specific type of instrument (Imspector) and so it is still justified for a publication despite of abundant of information already in the public domain. As far as the reviewer is concerned the strength of this paper may be the distortion assessments for this particular type of HSI line scanner, and the other information should be heavily condensed or even removed to improve the clarity of the paper : The details of the ‘Recorder’ software such as binning/interface gui etc should be all removed; The presentation of the experimental data collected by the system is really not necessary, or may be just one example is more than enough. Suggestions for further improvement: The paper should be focussed more on distortion assessments of HSI line scanners and so the title should reflect this and something like : ’Spectral & spatial distortions assessments in typical hyperspectral imaging line-scanner system’; Remove table 2; Remove the binning info in fig 3; Give more info about the IMU (line 160) particularly its spatial resolution/accuracy; Remove fig 5; Expand section 3.1 and to give more details of fig 8 (which is one of the main results of the paper). Also expand discussions on fig 9 & 10 if possible. About the experimental result it is suggested to use one example instead of two so to keep the paper more focussed on the distortion assessment. Questions about the vines data set: what is ‘medium value’ in fig 13? What is the implication of this figure? The discussion and conclusion is far too short: needs substantial expansion!  

Authors Coment:

Thank you for the detailed reading of the manuscript and positive comments.

This manuscript has been written with an intention to present and provide an overview of imaging spectrometer system the calibration method and its usefulness for many practical applications. Following your suggestion, the distortion estimates for this particular type of HSI line scanner are particularly emphasized in the text. In line with that change, we also changed the title of the paper.

However, we tried to answer all your comments and inquiries, and we did major revisions of the manuscript.

Particular remarks and suggestions for further improvement:

Ad1 - Reviewer: 2

As far as the reviewer is concerned the strength of this paper may be the distortion assessments for this particular type of HSI line scanner, and the other information should be heavily condensed or even removed to improve the clarity of the paper:

The details of the ‘Recorder’ software such as binning/interface gui etc should be all removed; The presentation of the experimental data collected by the system is really not necessary, or may be just one example is more than enough.

Ad1 - Authors Comments

We have reduced details of the ‘Recorder’ software such as binning/interface gui etc should be all removed; We presented just one example of the experimental data collected by the system.

Ad2 - Reviewer: 2

The paper should be focussed more on distortion assessments of HSI line scanners and so the title should reflect this and something like : ’Spectral & spatial distortions assessments in typical hyperspectral imaging line-scanner system’.

Ad2 - Authors Comments

The distortion estimates for this particular type of HSI line scanner are particularly emphasized in the text. In line with that change, we also changed the title of the paper.

Ad3 - Reviewer: 2

Remove table 2.

Ad3 - Authors Comments

It’s done.

Ad4 - Reviewer: 2

Remove the binning info in fig 3.

Ad4 - Authors Comments

It’s done.

Ad5 - Reviewer: 2

Give more info about the IMU (line 160) particularly its spatial resolution/accuracy.

Ad 5 - Authors Comments

It’s done.

Ad6 - Reviewer: 2

Remove fig 5.

Ad6 - Authors Comments

It’s done.

Ad7 - Reviewer: 2

Expand section 3.1 and to give more details of fig 8 (which is one of the main results of the paper). Also expand discussions on fig 9 & 10 if possible.

Ad7 - Authors Comments

We added some text in section 3.1, but also in section 2.3 and 4, which are linked with it.

Ad8 - Reviewer: 2

About the experimental result it is suggested to use one example instead of two so to keep the paper more focussed on the distortion assessment.

Ad8 - Authors Comments

We presented just one example of the experimental data collected by the system.

Ad9 - Reviewer: 2

The discussion and conclusion is far too short: needs substantial expansion!

Ad9 - Authors Comments

It’s done.

Yours sincerely,

Andrija Krtalić

Vanja Miljković

Dubravko Gajski

Ivan Racetin

Round 2

Reviewer 2 Report

The revised manuscript has addressed all the points raised in the previous review, however, the use of English in the present form of the paper is needed to improve. It is suggested to improve the readability of the paper through professional service or by an English speaking person.

Author Response

Response to Reviewer 2

____________________________________________________________________________

Reviewer: 2

Comments to the Author

The revised manuscript has addressed all the points raised in the previous review, however, the use of English in the present form of the paper is needed to improve. It is suggested to improve the readability of the paper through professional service or by an English speaking person.

Authors Coment:

Thank you, once more, for the detailed reading of the manuscript and positive comments.

We sent the manuscript for editing by a professional service of the MDPI. Below, you can find MDPIs verification of this.

Yours sincerely,

Andrija Krtalić

Vanja Miljković

Dubravko Gajski

Ivan Racetin
